# Clinical Relevance of Food Addiction in Higher Weight Patients across the Binge Eating Spectrum

**DOI:** 10.3390/bs14080645

**Published:** 2024-07-26

**Authors:** Alycia Jobin, Félicie Gingras, Juliette Beaupré, Maxime Legendre, Catherine Bégin

**Affiliations:** 1School of Psychology, Université Laval, Quebec City, QC G1V 0A6, Canada; alycia.jobin.1@ulaval.ca (A.J.); felicie.gingras.1@ulaval.ca (F.G.); juliette.beaupre.1@ulaval.ca (J.B.); maxime.legendre.1@ulaval.ca (M.L.); 2Centre d’Expertise Poids, Image et Alimentation (CEPIA), Université Laval, Quebec City, QC G1V 0A6, Canada

**Keywords:** binge eating disorder, eating behavior, food addiction, obesity

## Abstract

Food addiction (FA) is associated with greater severity on many eating-related correlates when comorbid with binge eating disorder (BED) but no study has established this relation across the whole spectrum of binge eating, i.e., from no BED to subthreshold BED to BED diagnosis. This study aims to examine the effect of the presence of FA on the severity of eating behaviors and psychological correlates in patients without BED, subthreshold BED or BED diagnosis. Participants (n = 223) were recruited at a university center specialized in obesity and eating disorder treatment and completed a semi-structured diagnostic interview and questionnaires measuring eating behaviors, emotional regulation, impulsivity, childhood interpersonal trauma, and personality traits. They were categorized by the presence of an eating disorder (no BED, subthreshold BED, or BED) and the presence of FA. Group comparisons showed that, in patients with BED, those with FA demonstrated higher disinhibition (*t*(79) = −2.19, *p* = 0.032) and more maladaptive emotional regulation strategies (*t*(43) = −2.37, *p* = 0.022) than participants without FA. In patients with subthreshold BED, those with FA demonstrated higher susceptibility to hunger (*t*(68) = −2.55, *p* = 0.013) and less cooperativeness (*t*(68) = 2.60, *p* = 0.012). In patients without BED, those with FA demonstrated higher disinhibition (*t*(70) = −3.15, *p* = 0.002), more maladaptive emotional regulation strategies (*t*(53) = −2.54, *p* = 0.014), more interpersonal trauma (*t*(69) = −2.41, *p* = 0.019), and less self-directedness (*t*(70) = 2.14, *p* = 0.036). We argue that the assessment of FA provides relevant information to complement eating disorder diagnoses. FA identifies a subgroup of patients showing higher severity on many eating-related correlates along the binge eating spectrum. It also allows targeting of patients without a formal eating disorder diagnosis who would still benefit from professional help.

## 1. Introduction

Food addiction (FA) postulates that some patients may lose control of their consumption due to an addiction to certain foods [1]. FA is measured by the Yale Food Addiction Scale 2.0 [2], a self-reported questionnaire based on the DSM-5 criteria for substance-related and addictive disorders [3]. Patients reporting FA experience symptoms like a feeling of losing control over food, irrepressible cravings, and a mood enhancement felt at the time of consumption [2]. While FA is not a recognized psychiatric disorder in the DSM-5 [3], it has still generated much interest in the field of eating disorders. Many studies have pointed out similarities and differences between FA and binge eating disorder (BED) [4,5]. Conversely, some authors have proposed that FA could represent a more severe form of BED embedded in a biological state of dependence [6,7,8]. In clinical terms, that would mean that, when treating patients with BED, those who would report FA would have a more complex and harder-to-treat condition.

A recent systematic review showed that FA is present concomitantly with BED in 42% to 92% of the time [9] and many studies demonstrated that this dual condition is associated with higher psychiatric comorbidities, cravings, shape and weight concerns, and psychological distress [10,11,12]. While this proposition of FA as a worsened condition of BED is very interesting, it cannot account for the high proportion of patients reporting FA without BED. In fact, many studies that investigated FA in patients with obesity without BED found that FA is also associated with higher eating, shape and weight concerns, higher frequency of objective binge eating episodes [13] and higher psychiatric comorbidities [14]. Given these results, FA may represent a clinical construct to understand a more severe form of BED, along with the difficulties experienced by patients who report a loss of control overeating without meeting full criteria for BED. To our knowledge, no study has looked at FA along the whole spectrum of binge eating including those with subthreshold BED as well as those without a formal diagnosis but presenting compulsive eating.

To better account for the results presented above, a more comprehensive explanation would be that FA is associated with higher severity of many eating-related correlates in every patient, even those without BED. First, a previous study showed that more disinhibition, more susceptibility to hunger and less restraint are generally found in BED compared to patients without BED [15]. Similarly, FA symptoms are positively correlated with disinhibition and susceptibility to hunger but not restraint [16,17]. Second, a systematic review showed that BED is associated with more maladaptive emotion regulation strategies [18], and poorer emotion regulation was also found in patients with FA [19,20]. A recent finding also suggests that emotion regulation would be particularly impaired in patients with BED and FA compared to patients with only FA [21]. Third, general impulsivity has been demonstrated to account for higher severity in a sample of patients with BED [22] and a recent systematic review established that many facets of impulsivity are consistently associated with FA [23]. Fourth, interpersonal trauma has been repeatedly associated with greater severity of BED [24] and FA symptoms [25,26,27]. Finally, some common personality trait patterns can be observed in patients with BED, such as low self-directedness, high harm avoidance, and high reward dependence [28,29], and in patients with FA, such as low self-directedness, low cooperation, high novelty seeking, and high harm avoidance [14,20,30].

To date, it is still unclear if those differences found between patients with and without FA would be found equally across the binge eating spectrum, from no BED, subthreshold BED to full BED diagnosis. In other words, the higher severity associated with the presence of FA in patients with BED will also be found in patients elsewhere on the binge eating spectrum. Using a binge eating spectrum, including three groups (no BED, subthreshold BED, and BED), the objective of this study was to explore if the presence of FA would allow differentiation of patients with a more complex condition within each group using the following markers: eating behaviors (restraint, disinhibition, and susceptibility to hunger), personality traits, impulsivity, emotion regulation and childhood interpersonal trauma. The hypothesis was that for each group (no BED, subthreshold BED, and BED), the presence of a FA diagnosis would be associated with some characteristics that complexify the condition and make it harder to treat.

## 2. Methods

### 2.1. Participants

Participants were recruited among patients seeking a psychologist at the Centre d’Expertise Poids, Image et Alimentation (CEPIA) specialized in difficulties related to weight, eating, or body image. To be included in the study, they had: (1) to be 18 years old, (2) to reach the overweight threshold (self-reported Body Mass Index (BMI) ≥ 25 kg/m^2^), and (3) to report a form of compulsive eating or loss of control. Patients with a diagnosis of anorexia nervosa, bulimia, pica, rumination disorder, or avoidant/restrictive food intake disorder were excluded. The sample consisted of 223 patients, 199 women (89.24%) and 24 men (10.76%), aged between 21 and 72 years (M = 45.17) and with a BMI ranging from 25.2 to 74.6 kg/m^2^ (M = 38.17). The majority of the sample identified as white. Half of the sample had an income of $80,000 or more, and around 60% of the participants had been to university and were married or in a common-law relationship (Table 1).

### 2.2. Procedure

Recruitment at the CEPIA has been ongoing from April 2019 to November 2023. Each patient completed questionnaires on an online platform (LimeSurvey) and a semi-structured EDs diagnostic interview based on the DSM-5 criteria conducted by a psychologist. Patients gave their consent to be anonymously included in a database used for the present study. The Laval University Research Ethics Committee approved the study.

### 2.3. Measures

Eating disorders were assessed by psychologists following a semi-structured interview and a checklist based on DSM-5 criteria. Each psychologist had to check off the endorsed criteria and make a clinical decision on the presence or absence of an eating disorder and its severity. A sociodemographic questionnaire was used to collect data from participants, such as age, gender, ethnicity, education, marital status, household income, height, and weight.

#### 2.3.1. Food Addiction (FA)

FA was assessed by the Yale Food Addiction Scale 2.0 (YFAS) [2,31]. This self-reported questionnaire includes 35 items to retrospectively assess the presence of 11 criteria in the past year using an 8-point Likert scale (ranging from 0 = never to 7 = every day). In order for a “diagnosis” of FA to be established, the participant must meet two of the 11 criteria in addition to the criterion for significant distress and/or impaired functioning. The severity of FA is determined by the number of criteria met (ranging from 0 to 11). According to previous studies, this scale has good internal consistency with a Kuder–Richardson coefficient varying between 0.83 and 0.84 [28,31].

#### 2.3.2. Restraint, Disinhibition, and Susceptibility to Hunger

The Three-Factor Eating Questionnaire (TFEQ) [32] was used to measure three forms of eating behavior: restraint, disinhibition, and susceptibility to hunger. It is a self-reported questionnaire with 51 items answered by true or false or using a 4-point Likert scale (ranging from 1 = low endorsement to 4 = strong endorsement). A total score can be generated for the three scales. The scale for restraint consists of 21 items (e.g., I often stop eating when I am not really full as a conscious means of limiting the amount that I eat), the one for disinhibition consists of 16 items (e.g., When I am with someone who is overeating, I usually overeat too), and the one for susceptibility to hunger consists of 14 items (e.g., Being with someone who is eating often makes me hungry enough to eat also). These subscales have good internal consistency with a Cronbach’s alpha ranging from 0.85 to 0.93 [33].

#### 2.3.3. Emotion Regulation

The Cognitive Emotion Regulation Questionnaire (CERQ) [34] was used to assess participants’ emotional regulation. This 36-item self-report questionnaire covers nine cognitive strategies that individuals might use when facing negative events, four of which are considered adaptive and five of which are considered maladaptive. The CERQ includes nine subscales, each with four items. The internal consistency of this questionnaire is acceptable, with a Cronbach’s alpha of 0.70 [35].

#### 2.3.4. Impulsivity

The short version of the Impulsive Behavior Scale (UPPS-P) [36] assesses five different facets of impulsivity: positive urgency, negative urgency, lack of premeditation, lack of persistence, and sensation seeking. Each of these facets includes four items that are rated on a Likert scale ranging from 1 = “I strongly agree” to 4 = “I strongly disagree”. This shortened version therefore contains a total of 20 items. The internal consistency of this scale is good with a Cronbach’s alpha varying between 0.70 and 0.84 depending on the subscale [37].

#### 2.3.5. Childhood Interpersonal Trauma

The Cumulative Childhood Trauma Questionnaire (CCTQ) [38], a self-report questionnaire, was used to measure childhood trauma. This questionnaire measures five different types of childhood interpersonal trauma: sexual abuse, physical abuse, emotional abuse, physical neglect, and emotional neglect. These are measured using different scales. In order to produce a cumulative trauma severity score, the severity scores of the five types of traumas are added together. The validity of the tool has been demonstrated through adequate internal consistency with a Cronbach’s alpha of 0.86 [27].

#### 2.3.6. Personality Traits

The Temperament and Character Inventory (TCI-125) [39] is a 125-item version of the longer 226-item self-administered questionnaire that assesses seven personality dimensions: novelty seeking, harm avoidance, reward dependence, persistence, self-directedness, self-transcendence, and cooperativeness. For each item, the respondent must indicate true or false. The internal consistency of this questionnaire is acceptable with Cronbach’s alpha between 0.63 and 0.87 depending on the scale [40].

### 2.4. Statistical Analysis

Statistical Package for the Social Sciences (SPSS) 24 software was used. Prior to analyses, all variables’ distributions were inspected, and it was determined that no transformation was needed. Eighty-seven participants did not complete the CERQ because this questionnaire was introduced later (n for CERQ = 136), sixty-five participants refused to provide height and/or weight (n for BMI = 158), and one participant refused to answer questions about interpersonal trauma (n for CCTQ = 222). Participants were separated into three groups based on the semi-structured EDs diagnostic interview (no BED, n = 72; subthreshold BED, n = 70; and BED, n = 81). The subthreshold BED group consisted of patients diagnosed with other specified feeding and eating disorders (OSFED) because they reported binge eating episodes without meeting the frequency or duration criteria. Firstly, each group were compared on sociodemographic characteristics using one-way ANOVA or Chi-squared test. Secondly, within each group, patients without FA and patients with FA were compared on maladaptive eating behaviors and psychological correlates, with *t* tests with Bonferroni correction for multiple comparisons.

## 3. Results

Overall, 66.37% of participants met the criteria for FA. More specifically, 81.48% of participants with BED, 65.71% of participants with subthreshold BED, and 50% of participants without BED met the criteria for FA. The average FA symptoms score for the sample was 5.20 (SD = 3.31). Patients without BED had significantly fewer FA symptoms (3.76) than patients with subthreshold BED (5.27) and BED (6.42), while the difference between patients with subthreshold BED and BED was only marginally significant (*p* = 0.077). In patients with BED, those with FA compared to those without FA demonstrated higher disinhibition (FA: 12.02, no FA: 10.53) and more maladaptive emotional regulation strategies (FA: 40.32, no FA: 34.73). In patients with subthreshold BED, those with FA compared to those without FA demonstrated higher susceptibility to hunger (FA: 7.71, no FA: 5.83) and less cooperativeness (FA: 81.74, no FA: 88.67). Finally, in patients without BED, those with FA compared to those without FA demonstrated higher disinhibition (FA: 10.94, no FA: 8.83), more maladaptive emotional regulation strategies (FA: 43.40, no FA: 36.56), more interpersonal trauma (FA: 17.94, no FA: 9.80), and less self-directedness (FA: 60.56, no FA: 71) (see Table 2).

## 4. Discussion

The aim of this study was to examine if the presence FA would be associated with some characteristics that complexify the condition of patients on the binge eating spectrum (no BED, subthreshold BED, and BED). At different levels, maladaptive eating behaviors (disinhibition and susceptibility to hunger), maladaptive emotion regulation strategies, impulsivity, and childhood interpersonal traumas were all higher in the presence of FA across the binge eating spectrum. First, for FA patients without BED or those with BED, significantly higher disinhibition was observed and, for FA patients with subthreshold BED, significantly higher susceptibility to hunger was observed. It is noteworthy that, in the presence of FA, means were consistently higher on both disinhibition and susceptibility to hunger across the entire spectrum. Consistent with the literature, these results support that FA is associated with more severe eating behaviors in both patients with and without BED [16,17] and that this association also applies to subthreshold BED patients. Second, a similar trend was found for maladaptive emotion regulation strategies, impulsivity, and childhood interpersonal traumas. In patients without BED, those with FA showed significantly higher maladaptive emotion regulation strategies and childhood interpersonal traumas than those without FA. In patients with BED, those with FA showed significantly higher maladaptive emotion regulation strategies than those without FA. While other differences were not significant, for maladaptive emotion regulation strategies, impulsivity, and childhood interpersonal traumas a consistent increase was found in the presence of FA across the whole spectrum of BED. Different facets of emotional regulation and impulsivity have been studied in relation to both BED [18,22] and FA [19,20,21,23] and, along with the present results, suggest that emotion regulation impairment and higher impulsivity are commonly associated with FA as well as with more severe clinical presentation. Finally, FA was associated with more childhood interpersonal traumas as shown by many previous studies that demonstrated either correlations between the number of FA symptoms and childhood trauma score or more frequent traumatic history in patients with FA [25,26,27].

For personality traits, the differences observed were much more marginal and only two significant differences were found. The presence of FA was associated with less self-directedness in patients without BED and with less cooperativeness in subthreshold BED patients, two personality traits that have been associated with FA previously [30]. Additionally, self-directedness and harm avoidance both showed trends in line with the idea that FA is associated with higher severity. Self-directedness was systematically lower in the presence of FA for each of the three groups and, inversely, harm avoidance was systematically higher in the presence of FA for each of the three groups. These two personality traits have previously been highlighted as prominent correlates of both BED [28,29] and FA in samples of patients with eating disorders or with severe obesity awaiting bariatric surgery [20,30]. In the present context, it suggests that patients with FA are those most likely to present difficulties in setting, focusing, and directing their efforts towards long-term goals (self-directedness) and to show difficulties in tolerating negative affect to which they may respond with avoidance strategies (harm avoidance). The exploration of personality traits in relation to FA needs to be pursued with a measure based on pathological personality traits to better isolate patients with higher severity.

A key point that is important to raise is that patients who seek clinical help without obtaining a formal BED diagnosis had high severity on several indicators. Mainly, the number of FA symptoms endorsed by these patients was 3.76, which is higher than the average of 2.38 found in a representative sample of the general population [2]. In that group specifically, patients with FA clearly showed a different severity profile that warrants clinical attention. Therefore, it suggests that FA assessment allows for the detection of eating-related distress that is not covered by the current DSM-5 eating disorder classification. The present results are not intended to challenge the DSM-5 classification or to advocate for the inclusion of FA in the DSM-5. Rather, it appears that the assessment of FA can rapidly target patients who experience eating difficulties different from binge eating or patients at risk of binge eating episodes. Specifically, these may be non-binge patients who have problems with emotional eating or grazing and who are very concerned about their shape and weight [41]. Future studies should focus on capturing the diversity of eating difficulties in patients with higher weight as it goes beyond binge eating episodes. It would be relevant to study the evolution over time of different eating-related correlates and food addiction symptoms on the binge eating spectrum.

This study has limitations to consider. First, although the total sample size is relatively large for the population studied (N = 223), the number of patients with BED without FA and with subthreshold BED without FA were small (n = 15 and 24, respectively). This certainly had a negative impact on statistical power, but the observation of the means for each group still makes it possible to appreciate the increase in severity according to the FA diagnosis. Second, while many important correlates were considered in the study, the presence of comorbid psychiatric disorders was not assessed. That would have been an important factor to control for. Thirdly, the sample in the present study consisted predominantly of women, Caucasian, married, and highly educated. Replication with a more diverse sample would be an important step for the generalization of these results. Finally, some participants (n = 65) decided not to provide their height and weight for personal reasons resulting in missing data for BMI.

## 5. Conclusions

The aim of this study was to examine if the presence of FA would be associated with some specific characteristics suggesting higher severity for patients with compulsive eating along BED spectrum from no BED, subthreshold BED to full BED diagnosis. The most important finding is that, no matter the level of BED (from none to full BED), the presence of FA was associated with more maladaptive eating behaviors (disinhibition and susceptibility to hunger), maladaptive emotion regulation strategies, impulsivity, and childhood interpersonal traumas. As suggested by previous studies [7,8], our results support that FA captures more severe clinical presentations in patients reporting loss of control and overeating. Secondly, the study showed that patients who seek clinical help for compulsive eating without obtaining a formal eating disorder diagnosis may still present a concerning level of maladaptive eating behaviors and emotional regulation difficulties. Particularly for these patients, FA assessment allows targeting of those with a more severe profile who should probably receive professional help at the same level as their colleagues with a formal eating disorder. FA appears to be helpful in capturing dimensions of eating pathology that are not captured by the DSM-5 and, therefore, to detect patients that would have been undetected after a semi-structured diagnostic evaluation. This study highlights the relevance of providing psychological help without restricting accessibility exclusively to those who meet the DSM-5 diagnostic criteria for an eating disorder.

## Figures and Tables

**Table 1 behavsci-14-00645-t001:** Sociodemographic characteristics according to the presence of BED.

	No BED (n = 72)	Subthreshold BED (n = 70)	BED (n = 81)	Anova/χ^2^
Mean age (SD)	45.45 (11.57)	48.13 (12.62)	42.22 (12.50)	*F* (2, 221) = 4.41, *p* = 0.013 *
Mean BMI (SD)	35.59 (7.00)	37.81 (6.97)	41.37 (10.00)	*F* (2, 156) = 7.30, *p* = 0.001 *
Gender (%)				χ^2^ = 1.13, *p* = 0.569
Women	90.28%	91.43%	86.42%	
Men	9.72%	8.57%	13.58%	
Household income (%)				χ^2^ = 7.03, *p* = 0.856
Less than $40,000	19.44%	20.00%	14.82%	
$40,000 to $79,999	22.22%	37.14%	38.27%	
$80,000 and more	48.62%	34.29%	35.80%	
Prefer not to answer	9.72%	8.57%	11.11%	
Education (%)				χ^2^ = 1.89, *p* = 0.756
High school	9.73%	11.43%	16.05%	
College	31.94%	32.86%	27.16%	
University	58.33%	55.71%	56.79%	
Marital status (%)				χ^2^ = 10.01, *p* = 0.264
Married/common-law	59.73%	52.86%	60.49%	
Divorced/separated	19.44%	10.00%	9.88%	
Widow(er)	0.00%	1.43%	1.23%	
Single	20.83%	35.71%	28.40%	
Ethnicity (%)				χ^2^ = 6.17, *p* = 0.404
White	98.61%	95.71%	97.53%	
Other	1.39%	4.29%	2.47%	

Note. BMI = body mass index; BED = binge eating disorder. * Significant results.

**Table 2 behavsci-14-00645-t002:** Means (standard deviation) and *t* test comparisons of the severity indicators according to the presence of food addiction for each eating disorder.

	No BED (n = 72)			Subthreshold BED (n = 70)			BED (n = 81)		
	No FA (n = 36)	FA (n = 36)	*t* Test	No FA (n = 24)	FA (n = 46)	*t* Test	No FA (n = 15)	FA (n = 66)	*t* Test
FA symptoms	1.61 (1.99)	5.92 (2.56)	*t*(70) = −7.97, *p* < 0.001	3.13 (2.59)	6.39 (2.53)	*t*(68) = −5.09, *p* < 0.001	2.47 (2.45)	7.32 (2.72)	*t*(79) = −6.35, *p* < 0.001
Eating behaviors									
Restraint	8.67 (4.49)	9.06 (4.22)	*t*(70) = −0.38, *p* = 0.706	9.38 (5.06)	8.04 (3.98)	*t*(68) = 1.12, *p* = 0.269	7.33 (4.73)	7.58 (4.42)	*t*(79) = −0.19, *p* = 0.850
Disinhibition	8.83 (2.54)	10.94 (3.12)	*t*(70) = −3.15, *p* = 0.002 ***	10.63 (2.02)	11.17 (1.83)	*t*(68) = −1.15, *p* = 0.254	10.53 (3.04)	12.02 (2.19)	*t*(79) = −2.19, *p* = 0.032 ***
Susceptibility to hunger	5.69 (3.55)	7.08 (4.00)	*t*(70) = −1.56, *p* = 0.124	5.83 (3.37)	7.91 (3.17)	*t*(68) = −2.55, *p* = 0.013 ***	8.27 (3.37)	8.86 (3.43)	*t*(79) = −0.61, *p* = 0.544
Personality traits									
Novelty seeking	42.64 (19.62)	41.53 (17.76)	*t*(70) = 0.25, *p* = 0.802	40.21 (18.27)	44.46 (19.70)	*t*(68) = −0.88, *p* = 0.383	39.00 (16.50)	43.18 (19.57)	*t*(79) = −0.77, *p* = 0.445
Harm avoidance	58.25 (26.44)	60.06 (25.23)	*t*(70) = −0.30, *p* = 0.768	57.96 (29.70)	59.87 (23.95)	*t*(68) = −0.29, *p* = 0.772	57.67 (30.35)	67.20 (22.57)	*t*(79) = −1.15, *p* = 0.267
Reward dependence	72.64 (16.03)	70.53 (17.76)	*t*(70) = 0.53, *p* = 0.598	72.83 (17.03)	70.39 (16.04)	*t*(68) = 0.59, *p* = 0.556	75.87 (13.45)	72.08 (14.61)	*t*(79) = 0.92, *p* = 0.361
Persistence	66.67 (25.30)	75.00 (24.55)	*t*(70) = −1.42, *p* = 0.161	75.00 (27.19)	66.96 (28.66)	*t*(68) = 1.13, *p* = 0.261	61.33 (35.02)	66.97 (32.25)	*t*(79) = −0.60, *p* = 0.549
Self-directedness	71.00 (19.42)	60.56 (21.89)	*t*(70) = 2.14, *p* = 0.036 ***	66.92 (19.53)	63.39 (19.25)	*t*(68) = 0.72, *p* = 0.472	67.20 (19.72)	58.55 (19.09)	*t*(79) = 1.65, *p* = 0.104
Self-transcendence	35.75 (20.88)	40.64 (27.02)	*t*(70) = −0.86, *p* = 0.393	38.50 (24.85)	33.24 (20.62)	*t*(68) = 0.94, *p* = 0.349	31.13 (18.90)	34.08 (23.42)	*t*(79) = −0.45, *p* = 0.651
Cooperativeness	86.00 (9.30)	86.89 (12.42)	*t*(70) = −0.34, *p* = 0.732	88.67 (9.11)	81.74 (12.97)	*t*(68) = 2.60, *p* = 0.012 *	86.93 (10.31)	83.58 (11.25)	*t*(79) = 1.06, *p* = 0.293
Impulsivity	43.11 (7.08)	44.47 (6.82)	*t*(70) = −0.83, *p* = 0.409	41.17 (7.76)	43.41 (7.85)	*t*(68) = −1.14, *p* = 0.258	44.47 (7.76)	46.20 (7.95)	*t*(79) = −0.76, *p* = 0.447
Emotional regulation									
Adaptative regulation	60.44 (11.63)	63.13 (15.99)	*t*(53) = −0.70, *p* = 0.486	59.45 (10.51)	58.00 (14.30)	*t*(34) = 0.30, *p* = 0.764	58.45 (14.64)	57.68 (10.42)	*t*(43) = 0.194, *p* = 0.847
Maladaptive regulation	36.56 (7.42)	43.40 (12.32)	*t*(53) = −2.54, *p* = 0.014 ***	37.09 (8.77)	38.60 (10.19)	*t*(34) = −0.43, *p* = 0.673	34.73 (7.25)	40.32 (6.66)	*t*(43) = −2.37, *p* = 0.022 *
Interpersonal traumas	9.80 (10.27)	17.94 (17.39)	*t*(69) = −2.41, *p* = 0.019 ***	12.42 (12.85)	14.93 (12.44)	*t*(68) = −0.80, *p* = 0.429	11.53 (17.68)	14.42 (12.93)	*t*(79) = −0.73, *p* = 0.469

Note. FA = food addiction; BED = binge eating disorder. * Significant results.

## Data Availability

The datasets used and/or analyzed during the current study are available from the corresponding author on reasonable request.

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
