# Peer review of "Clinical Relevance of Food Addiction in Higher Weight Patients across the Binge Eating Spectrum"

_behavsci, 2024, doi:10.3390/bs14080645_

Round 1

Reviewer 1 Report

Comments and Suggestions for Authors

I appreciate the opportunity to review this manuscript and congratulate the authors for their research. Below, I describe some suggestions to make the paper even better.

Summary: Enter numeric data

Formatting: I suggest using numerical citation.

Introduction: Break up the paragraphs, because they are too long. The introduction was very long, some information can be used in the discussion of the results, which was brief.

Method: Were participants evaluated by a psychiatrist to confirm the diagnosis of BED or AF? If not, I suggest that the authors treat them as "behaviors suggestive" of BED and FA.

Results: Describe table 1 and table 4

Conclusion: I suggest leaving suggestions for future studies in the discussion section and not in the conclusion.

Reviewer 2 Report

Comments and Suggestions for Authors

Review Report for: “Food addiction to capture severe condition of eating disorder”

Summary

The study is focused on utilizing a number of survey instruments to determine if the severity of BED or EDNOS is captured appropriately with the diagnosis. It also examines if food addiction should be used to support clinical care for individuals regardless of ED dx.

General Comments

·       You generally mention the term Eating Disorders but the study is only focused on two that are in the DSM 5 (BED and OSFED). EDNOS is no longer used and it now OSFED (Other Specified Feeding and Eating Disorders). Other ED in the DSM 5 include ARFID, BN, AN, pica, and rumination disorder. Because you reference the DSM 5 and AN and BN in the methods (lines 116-117), I would mention the exclusion of these others as well.

·       I struggle to see how the personality traits, impulsivity, and emotional regulation can be used to distinguish severity – with the description of the tool (TCI) it seems these are general personality areas vs. actual behaviors or any behaviors related specifically to food. Furthermore, it isn’t clear how increased severity of the ED was determined for BED or EDNOS using this tool.  I see there were correlations between personality traits but I didn’t see how that was associated with severity of the ED. The FA scoring tool does offer a level of severity for FA and that would make sense to use the validated levels of severity for FA.

·       Overall there are some interesting findings; however, the argument is lacking in severity of the ED vs. how can the FA tool be helpful in identifying patients who need additional care and what type of care they need.

Specific Comments

1 1.     The title is misleading because this article addresses only BED and EDNOS rather than EDs. It also implies that FA can determine severity of BED or EDNOS but that wasn’t really clear in the findings. The finding that stands out is that a dx of FA warrants clinical treatment as it closely aligns with EDNOS and BED and likely requires professional support in the treatment.

2.       Determine if you want to use EDNOS or OSFED. OSFED is in the DSM 5 and EDNOS is not.

3.       Reorganizing the introduction might help the reader get a better sense of what the article is going to address. It starts with really specific info to BED and EDNOS which may be more relevant later in the introduction, but what is really important is lines 54-61.

4.       Lines 76-90 identify the correlation to BED and personalities, but I don’t see any research connecting it back to the severity of the symptoms or intensity of cravings. Is there research to support the argument that you can use personality traits as predictors of severity in BED or EDNOS?

5.       Line 113 – you mention that patients came seeking a “professional” what does that mean (MD, nurse, dietitian, others?) Even without saying the name of the Centre – a general description of the type of work would be helpful to the reader. Is this a location that specializes in ED? If yes, that would be important for the limitations of the study. Do you have any information on how participants found the Centre?

6.       Line 115 you say that BMI was a requirement for participation, however, in line 190 you mention that only 158 provided height and weight for the BMI. Did you exclude those that did not provide BMI data? Adjusting the language in either place so it is clear to the reader how you determined eligibility for the study.

7.       Line 116-117 you mention AN and BN for exclusion criteria – did you also exclude pica, rumination disorder, ARFID? How were the dx determined – self report or dx at the Centre?

8.       Provide information on how BED and EDNOS dx was determined for the statistical analysis? Was it self-reported or did they get dx at the Centre? Who did the dx – was it required to be a medical dx or could someone “self-dx?”

9.       Line 202 – provide a little more narrative on what “different in the number of FA symptoms” means, statistically significant?

10.   Table 3 line 227 – include the (n) for the No FA vs. FA in the three categories.

11.   The argument for connecting the severity with the tools used is needed – I only see correlations being drawn between the very few personality traits that were significantly connected. I don’t see how that resulted in a connection to the severity of BED or EDNOS. It’s possible that I don’t fully understand how the analysis was completed – did you compare the personalities with the eating behaviors? Add in more analysis details to the methods as to how you correlated the ED, FA, and eating behaviors and psychological correlates (lines 192-197). Making the connection in the discussion would also be helpful.

12.   Lines 242-244 – same comment – how was FA associated with severity of BED or EDNOS? I understand that FA has its own severity categories but very few who were dx with BED or EDNOS did not have FA. So is it that BED and EDNOS doesn’t capture the severity or is the argument that many people with FA should be further screened for BED or EDNOS, and/or that they need additional clinical treatment for FA?

13.   Line 270-272 is the argument that maladaptive behaviors be included in the DSM-5? If the eating behaviors are severe, wouldn’t they be captured in the EDNOS or is this a different psychological treatment needed for maladaptive eating behaviors – is it related to the trauma experienced and therefore trauma treatment is indicated vs. eating disorder treatment? Articulating the argument more clearly is needed.

Comments on the Quality of English Language

Overall the English is good; however, a number of places where the plural is incorrect. 

Round 2

Reviewer 2 Report

Comments and Suggestions for Authors

The revisions are well done and have created a manuscript with more clarity! As the reader, I understand how the analysis supports the objectives and the literature review aligns well with the findings. The argument is clear - that FA makes BED harder to treat and more complex, alongside the personality traits. 

Lines 170, 193, 197, and 257 – OSFED is spelled incorrectly

Comments on the Quality of English Language

There are minor English issues throughout the entire manuscript, needs editing throughout. 

Author Response

We have made the requested changes to the quality of the English and corrected a few typos.